



# Hydrography and Circulation West of Sardinia in June 2014

Michaela Knoll[1], Ines Borrione[2], Heinz-Volker Fiekas[1], Andreas Funk[1], Michael P. Hemming[3], Jan Kaiser[3], Reiner Onken[4], Bastien Queste[3], and Aniello Russo[2]

[1]Wehrtechnische Dienststelle für Schiffe und Marinewaffen, Maritime Technologie und Forschung (WTD71), Berliner Straße 115, 24340 Eckernförde, Germany
[2]NATO Science and Technology Organization, Centre for Maritime Research and Experimentation (CMRE), Viale San Bartolomeo 400, 19126 La Spezia, Italy
[3]Centre for Ocean and Atmospheric Sciences (COAS), School of Environmental Sciences, University of East Anglia (UEA), Norwich Research Park, Norwich, NR4 7TJ, United Kingdom
[4]Helmholtz-Zentrum Geesthacht, Centre for Materials and Coastal Research (HZG), Max-Planck-Straße 1, 21502 Geesthacht, Germany

*Correspondence to:* michaelaknoll@bundeswehr.org

**Abstract.** In the mainframe of the REP14-MED sea trial in June 2014, the hydrography and circulation west of Sardinia, observed by means of gliders, shipborne CTD instruments, towed devices, and vessel-mounted ADCPs, are presented and compared with previous knowledge. So far, the circulation is not well known in this area, and the hydrography is subject to long-term changes. Potential temperature, salinity, and potential density ranges, as well as core values of the observed water masses were determined. Modified Atlantic Water (MAW), with potential density anomalies below 28.72 kg m$^{-3}$, showed a salinity minimum of 37.93 at 50 dbar. Levantine Intermediate Water (LIW), with a salinity maximum of about 38.70 at 400 dbar, was observed within a range of $28.72 < \sigma_\Theta$ [kg m$^{-3}$] $< 29.10$. MAW and LIW showed slightly higher salinities than previous investigations. During the trial, LIW covered the whole area from the Sardinian shelf to 7°15' E. Only north of 40° N was it tied to the continental slope. Within the MAW, a cold and saline anticyclonic eddy was observed in the southern trial area. The strongest variability in temperature and salinity appeared around this eddy, and in the southwestern part of the domain, where unusually low saline surface water entered the area towards the end of the experiment. An anticyclonic eddy of Winter Intermediate Water was recorded moving northward at 0.014 m s$^{-1}$. Geostrophic currents and water mass transports calculated across zonal and meridional transects showed a good agreement with vessel-mounted ADCP measurements. Within the MAW, northward currents were observed over the shelf and offshore, while a southward transport of about 1.5 Sv occurred over the slope. A net northward transport of 0.38 Sv across the southern transect decreased to zero in the north. Within the LIW, northward transport of 0.6 Sv across the southern transects were mainly observed offshore, and decreased to 0.3 Sv in the north where they were primarily located over the slope. This presentation of the REP14-MED observations helps to further understand the long-term evolution of hydrography and circulation in the Western Mediterranean, where considerable changes occurred after the Eastern Mediterranean Transient and the Western Mediterranean Transition.





## 1 Introduction

From 6 to 25 June 2014, an extensive experiment called REP14-MED was carried out west of Sardinia in the area between 39°12' N and 40°12' N and between 7°15' E and 8°18' E (Onken et al., 2017). In this paper, all dates refer to the year 2014 unless otherwise stated. The two research vessels *Alliance* and *Planet*, and a variety of measuring platforms were used to

capture the hydrography and circulation in the area. One of the objectives was to collect data for operational ocean forecasting, model validation, and the evaluation of forecasting skill. Other aims were to explore the variability of the ocean and to study mesoscale and sub-mesoscale features. For those investigations, a detailed analysis of the recorded measurements is necessary. Therefore, the aim of this study is to provide an overview of the hydrography and circulation west of Sardinia observed during REP14-MED. Within the trial period of 18 days, measurements were made at the same

position on several occasions. This enabled a study of short-term variations in the upper water column, The REP14-MED observations were compared with previous knowledge to investigate long-term changes.

The hydrography of the Sardinian Sea, which is located between 38° N and 42° N, and between 7° E and the Sardinian coast (Ribotti et al., 2004), is characterized by various water masses: Modified Atlantic Water (MAW; see Table 1 for list of

acronyms), Winter Intermediate Water (WIW), Temperature Minimum Layer (TML), Levantine Intermediate Water (LIW), Western Mediterranean Deep Water (WMDW), and Bottom Water (BW). Atlantic Water entering the Western Mediterranean (WMED) at the surface through the Strait of Gibraltar is intensively mixed and modified on its way towards the east. This water mass, referred to as MAW, covers approximately the upper 150 m of the WMED with a salinity range of 36.5 to 38.0 in the Algerian Basin (Benzohra and Millot, 1995) and 37.2 to 38.2 in the Sardinian Sea (Sorgente et al., 2003).

The core of MAW is characterized by a minimum in salinity found during summer at about 50-70 m depth and during winter close to the surface (La Violette, 1994). Based on historical data, the mean potential temperature and salinity of the core in summer in the Sardinian Sea are 17.71 °C and 37.61 with a standard deviation of 2.87 °C and 0.33, respectively (Sparnocchia et al., 1994). WIW consists of cooled and homogenized MAW formed in winter during cold wind events in the northern WMED (Juza et al., 2013). It is characterized by local minima in potential temperature and salinity at about 150 to

200 m depth, with magnitudes depending on the formation conditions. Values of 12.35 °C and 38.3 are observed in the zone of formation (Benzohra and Millot, 1995, Vargas-Yáñez et al., 2012). WIW mainly follows the path of MAW westward along the European continental slope and across the Algerian Basin (Millot, 1999; Benzohra and Millot, 1995). An additional pathway of WIW directly connecting the formation area to the Algerian Basin is expected to be induced by frontal eddies associated with the North Balearic Front (Millot, 1999; Fuda et al., 2000). This North Balearic front separates the

Atlantic Water reservoir in the south from the saltier and denser waters in the north (Olita et al., 2013). TML is observed in the Alboran Sea (Allen et al., 2008), the Algerian Basin (Benzohra and Millot, 1995) and the Sardinian Sea (Sorgente et al., 2003). It is not an autonomous water mass, but consists of WIW modified continually on its way through the WMED. Potential temperature and salinity values are below 13.2 °C and 38.3, respectively. LIW is formed in the eastern



Mediterranean, entering the WMED through the Strait of Sicily. It is characterized by high salinities with core values of about 38.7 (Juza et al., 2013; Juza et al., 2015; Bosse et al., 2015). In the Sardinian Sea, the mean core values of potential temperature and salinity during summer based on historical data are 13.51 °C and 38.58 with a standard deviation of 0.20 °C and 0.06, respectively (Sparnocchia et al., 1994). WMDW is formed in winter by deep convection of mixed MAW and LIW

in the northwestern corner of the WMED, mainly in the Gulf of Lions, with potential temperatures and salinities of 12.8 °C to 12.9 °C and 38.4 to 38.5, respectively. WMDW spreads through the WMED and occupies the water column below 800 m (La Violette, 1994; Schott et al., 1996; Rixen et al., 2005; Bosse et al., 2015).

In recent decades, water masses in the Mediterranean Sea have been subject to considerable change. The Eastern

Mediterranean Transient (EMT) caused a significant increase in the heat and salt content of the LIW (Roether et al., 2007; Roether et al., 2014). This increase in temperature and salinity has reached the western basins and has affected, amongst others, the characteristics of WMDW. Below 200 dbar, the mean temperature and salinity in the WMED have risen by about 0.04 °C and 0.015, respectively, per decade since 1961, with increases in LIW twice as large as in the deep water (Borghini et al., 2014). Since the 1950s, the heat and salt content of BW have increased steadily. This increase has heavily intensified

since 1985 (Rixen et al., 2005; Krahmann and Schott, 1998). A major deep water formation event in winter 2004/05 started the Western Mediterranean Transition (WMT) (Schroeder et al., 2016). Within two years, BW experienced an increase in potential temperature and salinity of 0.038 °C and 0.016, respectively (Schroeder et al., 2008). One reason for the changes in the deep water is the transfer of the EMT signal to the WMED, with higher heat and salt content of the advected LIW which is part of the WMDW formation.

The circulation in the Sardinian Sea is scarcely known from previous investigations. The upper 300 m of the water column is dominated by meso- (10-100 km) to sub-mesoscale (< 10 km) features, predominantly anticyclonic eddies of different origin (Puillat et al., 2003; Ribotti et al., 2004). Large Algerian Eddies (AE), with a diameter of 50 to 250 km, are generated in the Algerian Basin by baroclinic instabilities of the Algerian Current, which flows eastward along the African coast. These AEs

embed less saline water of Atlantic origin and circulate cyclonically within a gyre around the Algerian Basin with a mean speed of a few centimeter per second. They have lifetimes of up to 3 years (Puillat et al., 2002), often extending from the surface to about 350 m or deeper, and bring relatively fresh MAW from the Algerian coast to the Sardinian Sea (Millot, 1999; Testor and Gascard, 2005; Olita et al., 2013). Several AEs were observed in the Sardinian Sea between 7°24' E and 8° E during the MedGOOS experiments in 2001 and 2002, (Ribotti et al., 2004). Results of ocean general circulation models

show that these eddies feed the Western Sardinian Current (WSC), a quasi-permanent southward flow at the surface west of Sardinia (Olita et al., 2013; Olita et al., 2015; Pinardi et al., 2013). At the southwest corner of Sardinia, the WSC is located closer to the coast and reaches its maximum intensity. Coastal upwelling, especially evident in the southwestern part of Sardinia, is pre-conditioned by the WSC (Olita et al., 2013).




According to Ekman theory, northerly winds induce favorable conditions for upwelling along the western Sardinian coast. This could be observed during MedGOOS 3 in September 2001 where wind speeds between 4 and 12 m s$^{-1}$ were recorded (Ribotti et al., 2004). At that time, a vertical salinity inversion occurred at about 30 to 50 m depth in the whole area. Therefore, upwelling was leading to lower salinities close to the shore. This salinity inversion may have been generated by

evaporation or by saline water from the shelf pushed by the wind-induced Ekman transport further offshore. Water on the shelf can be significantly more saline than further offshore where AEs bring fresher water from the south (Ribotti et al., 2004).

The LIW spreads into the WMED between 200 and 800 m depth (La Violette, 1994). South of Sardinia, the LIW vein

flowing westward is about 50 km wide, with a core at about 300 m. At the southwest corner of Sardinia, it turns northward and follows the continental slope (Sorgente et al., 2003). The LIW vein becomes wider, shallower and cooler (Millot, 1999). It continues northward west of Corsica and turns to the west, flowing as a coastal current along the European slope, where it takes part in the formation of WMDW. At the southwest corner of Sardinia, the LIW sporadically spreads westward. Instead of a permanent flow branch, it is due to filaments of LIW swirling around AEs (Millot and Taupier-Letage, 2005) or due to

the formation of barotropic Sardinian Eddies (SE) with LIW characteristics (Testor and Gascard, 2005; Bosse et al., 2015). The SEs stay south of the North Balearic Front and are often entrained southward by the Algerian Gyre. SEs are formed at intermediate depths, but usually show a surface signature after some weeks (Olita et al., 2013). The westward transport of LIW into the WMED is estimated to be roughly 1 Sv (Millot, 1999), while the eastward transport of MAW at the Sardinian Channel is nearly 2 Sv. A rough estimation of the total northward transport between Sardinia and Mallorca (Spain) based on

water and salt budgets yielded about 0.4 Sv (Bethoux, 1980).

This paper is organized into six sections. Since several instruments were involved in this study, a comparison is carried out in Sect. 2 to distinguish between variations due to instrumental bias and due to environmental features. The characteristics of the different water masses recorded in the trial area are specified in Sect. 3. These are compared with previous investigations

and potential temporal changes are discussed. The vertical and horizontal distributions of the water masses are indicated and special features like eddies and fronts are presented. Based on the CTD profiles, taken at the same position on several occasions within the measuring period of 18 days, short-term variations in the upper water column are discussed in Sect. 4. Direct current observations obtained with vessel-mounted ADCPs are presented in Sect. 5 and compared with geostrophic velocities and transports calculated within the MAW and the LIW based on the CTD transects covering the area between the

Sardinian coast and the deep ocean at 7°15' E. Finally, the paper's findings are summarized in Sect. 6.



## 2 Data Set and Sensor Comparison

For this paper, temperature and salinity measurements were utilised from the following instruments: Sea-Bird CTDs from *Alliance* and *Planet*, an Oceanscience Underway CTD (UCTD), a towed CTD chain, a towed ScanFish, three Seagliders named Fin (SG537), Kong (SG524), and Minke (SG510), and six Slocum gliders called Elettra, Noa, Jade, Zoe, Dora, and

WTD71. The CTD data were mainly obtained along 11 zonal transects, partly repeated several times and spaced by 6 nmi (nautical miles) or about 11 km from each other. Further zonal transects situated in-between were surveyed several times back and forth by the gliders (Fig. 1).

A comparison between the different CTD instruments was carried out to ensure that observed differences were based on

actual environmental variability. In the deep ocean, where the natural variability of potential density anomaly ($\sigma_\Theta$) was low, a systematic difference of 0.008 kg m$^{-3}$ was observed between the CTD sensors of *Planet* and *Alliance* (Fig. 2), which was slightly above the expected accuracy of about 0.004 kg m$^{-3}$. Investigations of profiles deeper than 2000 dbar showed that this discrepancy was due to about 0.001 S m$^{-1}$ higher conductivity values recorded by *Planet*, but the cause of this bias is unknown (Knoll et al., 2015a, Onken et al., 2017). Water samples taken by *Planet*, and analysed by a Guildline Autosal

salinometer perfectly agreed with the corresponding CTD values, and the double sensors of the CTD probe of *Alliance* showed no significant differences. The Sea-Bird CTD sensors were calibrated in a similar fashion shortly before the cruise at the NATO Science and Technology Organization, Centre for Maritime Research and Experimentation (STO CMRE). Data obtained by the CTD instruments of both ships were processed with the same manufacturer software using almost identical parameters.

Systematic differences in $\sigma_\Theta$ in the deep ocean were also observed between the shipborne CTDs and some of the deep profiling gliders. Compared to the *Planet* CTD measurements, gliders Noa, Jade, and Minke recorded slightly lower $\sigma_\Theta$ by about -0.012, -0.008 and -0.003 kg m$^{-3}$, respectively, while glider Fin observed slightly higher $\sigma_\Theta$ of +0.006 kg m$^{-3}$ (Fig. 3). No significant difference was observed between the *Planet* measurements and glider Kong. Glider Dora exhibited high

variations in $\sigma_\Theta$ at about 900 dbar, whereas all other CTD instruments showed almost constant zonal values. Since measurements of *Planet* covered all latitudes of the trial area, the shifts cannot simply be explained by latitudinal changes. The differences in $\sigma_\Theta$ between the sensors of the shipborne CTDs and the gliders were small, but mostly constant. In contrast to $\sigma_\Theta$, variations in potential temperature and salinity at a pressure of 900 dbar were much higher, with values rising towards the east (Fig. 3). Due to the variability in potential temperature and salinity, the differences between $\sigma_\Theta$ values of the

different instruments cannot be ascribed to a specific sensor. Only the discrepancy between the Sea-Bird CTDs of both ships could be explained by differences in conductivity due to the existence of profiles below 2000 dbar (Knoll et al., 2015a).



Data retrieved from ARGO float 6901836, comprising of one deep profile down to a pressure of more than 1200 dbar in the trial area in June, agreed well with $\sigma_\Theta$ values obtained by *Alliance*, but a difference of about 0.008 kg m$^{-3}$ was seen between the float and *Planet* measurements (Fig. 4). Further casts were obtained using an UCTD at the outer limits of the trial area (Fig. 1). Some of these profiles reached pressure levels of more than 500 dbar. The data were processed by adding observed

pressure offsets and applying time constants to reduce salinity spikes. The $\sigma_\Theta$ values below 400 dbar were often about 0.015 kg m$^{-3}$ lower than the ones observed in nearby CTD casts of *Planet* (Fig. 5). However, the expected temperature and salinity accuracy of the UCTD was only 0.005 °C and 0.05, respectively. Therefore, the observed difference was not significant.

In the upper water column, the range of natural variations was too large to detect small differences between the different sensors. The measurements of the shallow gliders Elettra, Zoe and WTD71 were within the range of the CTD observations obtained by both ships. The same is true for the measurements obtained by the towed CTD chain and the ScanFish, recorded concurrently between 21 and 23 June. Both devices covered at most the upper 200 dbar of the water column, where the natural variability was high. Before towing, in-situ calibration coefficients of the CTD chain sensors were determined based

on simultaneously recorded CTD casts of *Planet*. Thus, no systematic discrepancy existed between the CTD chain and the Sea-Bird CTD of *Planet*. The accuracies of the CTD chain measurements were about 0.02 °C and 0.05, respectively. A comparison between the CTD chain and the ScanFish towed in sync showed no significant differences.

Concerning investigations of temperature and salinity variability in the upper water column, the differences between the
platforms were negligible when compared with natural variations. However, for studying changes in $\sigma_\Theta$ at greater depths, the differences between the different sensors have to be taken into account.

### 3 Water Masses

The characteristics of the different water masses, observed by means of all CTD instruments and gliders used during REP14-MED, are specified below and compared with earlier results (Table 2). The indicated salinity and potential density anomaly
values given in Table 2 correspond to the CTD observations of *Planet*. According to the sensor comparison results, these values exhibit uncertainties in conductivity and potential density of about 0.001 S m$^{-1}$ and 0.01 kg m$^{-3}$, respectively.

### MAW

The potential temperature and salinity of MAW in the trial area varied between 13.6 and 24 °C and between 37.1 and 38.3, respectively (Fig. 6a). The observed salinity range was slightly higher than the older measurements of Sparnocchia et al.
(1994) and Benzohra and Millot (1995) (Table 2). The salinity and $\sigma_\Theta$ values increased towards the shelf (Fig. 7). A local salinity minimum occurred at about 50 dbar, with mean values of 37.93 somewhat higher, and with about 15.19 °C



considerably colder, than the mean historical values observed by Sparnocchia et al. (1994). In order to understand how salinity within the core of the MAW changed with longitude, values and pressure levels of the salinity minima identified in all CTD profiles collected from *Planet* and *Alliance* are shown in Fig. 8 panels a and b, with sampling locations marked in Fig. 1. Close to the coast, the salinity minima within pressure values of 40 and 80 dbar increased to > 38.

The core of the MAW with the salinity minimum was located at about $\sigma_\Theta$=28.19 kg m$^{-3}$ (Fig. 6 a). The lower boundary was defined at the $\sigma_\Theta$=28.72 kg m$^{-3}$ isopycnal, which corresponded to a pressure level of 80 to 140 dbar (Fig. 8 c). Below this isopycnal, the salinity strongly increased as a result of LIW. Furthermore, TML and WIW were observed at depths below this density level.

Towards the end of the experiment, unusually low salinities of less than 37.2 occurred at the surface in the southwestern part of the trial area (Fig. 8 d). This feature did not appear in the earlier CTD measurements of *Planet* obtained in the same area at the beginning of the experiment. Similar salinity values of less than 37.2 were observed in AEs in the eastern part of the Algerian Basin (Benzohra and Millot, 1995, Taupier-Letage et al., 2003), and they also occurred in two anticyclonic eddies

in the Sardinian Sea (Sorgente et al., 2003).

Within the MAW, an intra-pycnocline anticyclonic eddy with a diameter of about 50 km was observed in the southern part of the domain, around 39°20' N and 7°43' E (Fig. 9). It was also apparent in transect 10, where the distance between the 27.5 and 28 kg m$^{-3}$ isopycnals strongly increased (Fig. 7 c, f, i). The presence of the eddy led to significantly modified local water

properties, with strongest effects around 50 dbar (Russo et al. 2015, Borrione et al., 2015). Salinities measured within the eddy were close to 38.2, and thus higher than those in the surrounding waters, in particular those to the west, where salinity ranged between 37.7 and 37.9 (Fig. 9). In the same area, just west of this eddy, where the low salinities were observed, the unusually low saline water mass described above, with salinities of less than 37.2, appeared at the surface at the end of the experiment.

**WIW/TML**

During REP14-MED, WIW was not observed as a continuous layer, but was confined to an anticyclonic WIW eddy detected by various platforms between 170 and 400 dbar (Russo et al., 2015). It had a diameter of about 11 km. Zonal transect 5 crossed the center of this eddy (Fig. 7 b, e, h). Several repeat CTD stations showed that the eddy moved northward during the experiment with a mean velocity of about 0.014 m s$^{-1}$. The eddy was characterized by potential temperatures below

13.1 °C and an almost constant $\sigma_\Theta$ value of 28.99 kg m$^{-3}$. The core values of potential temperature and salinity were < 12.9 °C and < 38.3, respectively, at about 250 dbar. These values were in agreement with previous observations of WIW in the northern WMED (Vargas-Yáñez et al., 2012; Juza et al., 2013), the Alboran Sea (Allen et al, 2008), and the Algerian





Basin (Benzohra and Millot, 1995) (Table 2), but to our knowledge no WIW eddy with such low temperatures and salinities has been documented in the Sardinian Sea in the past.

A relatively cold (< 13.3 °C) and fresh (< 38.3) water mass with core densities between 28.86 and 28.89 kg m$^{-3}$ was detected
between 120 and 270 dbar in the southwestern part of the trial area (Fig. 10). To distinguish this water mass from the WIW transported by the eddy, it is called TML, though it was strongly influenced by WIW. Since the observed TML was warmer and less dense than the WIW eddy, it is likely that the two water masses have a different history, having formed at different times. Several patches of TML water with slightly higher potential temperatures were observed during the MedGOOS experiments all over the Sardinian Sea (Sorgente et al., 2003, Ribotti et al., 2004).

Like the unusually low saline surface water, the TML entered the domain from the south and moved northward as indicated by the detection times (Fig. 10). It was not observed in the CTD measurements of *Planet*, which were obtained in the same area, but some time earlier before the first TML recording on the 18 June. Although TML and the unusually low surface salinities were detected in the same area at the end of the trial period, they did not always appear simultaneously as indicated
by the WTD71 glider measurements. While the TML was already observed for the first time on 18 June (Fig. 10) the unusually low surface salinities were recorded some time later on 22 June (Fig. 8 d).

**LIW**

The strong increase in salinity observed below $\sigma_\Theta$=28.72 kg m$^{-3}$ indicated the presence of the LIW layer (Fig. 6). Within the whole trial area, the mean potential temperature, salinity and $\sigma_\Theta$ values in the core of the LIW were 13.91 °C, 38.70 and
29.06 kg m$^{-3}$, respectively, at a mean pressure level of about 400 dbar (Fig. 11 a, b). Like MAW, LIW exhibited slightly higher salinity values compared to the historical data in the Sardinian Sea (Sparnocchia et al., 1994), likely due to the transfer of the EMT signal to the WMED (Roether et al., 2014).

East of 7°47' E, the LIW core salinity of 38.71 showed only small changes, while farther offshore it varied between 38.55
and 38.74 due to relatively low values in the northeast (Fig. 11 a, b). In contrast to earlier investigations, where LIW was attached closely to the Sardinian shelf break east of the 2000 m isobath (Puillat et al., 2003; Sorgente et al., 2003), during REP14-MED it covered the whole trial area west of the shelf. Only north of 40° N the LIW core was tied to the continental slope (Fig. 11 d). However, the thickness of the LIW layer decreased towards the west to less than 500 m due to the rise of the isopycnals (Fig. 7 g, h, i). For this study, the lower boundary of LIW was set to the potential density anomaly of
$\sigma_\Theta$=29.1 kg m$^{-3}$, which corresponded to a pressure level between 540 and 740 dbar (Fig. 11 c). Deeper than this, the vertical increase in $\sigma_\Theta$ was strongly reduced (Fig. 2).



**WMDW**

The REP14-MED data set showed that the WMDW was separated into an upper and lower part (Fig. 6). The upper part was strongly influenced by the overlying LIW. The potential temperature and salinity decreased with depth from 13.70 °C to 12.91 °C, and from 38.68 to 38.48, respectively, while $\sigma_\Theta$ slightly increased from 29.10 to 29.11 kg m$^{-3}$ (Fig. 2). Changes in

the lower part, found below approximately 1500 m, were very small. The potential temperature decreased, and salinity increased only slightly by about 0.01 °C and 0.005, respectively. These low values of about 12.90 °C and 38.48 were in agreement with WMDW specifications given in the literature (La Violette, 1994; Benzohra and Millot, 1995; Testor and Gascard, 2005). A further increase in potential temperature and salinity below 2300 dbar indicated the occurrence of BW.

**BW**

BW was observed at the deep CTD stations on the western side of the domain. Below 2300 dbar, the WMDW showed an increase in potential temperature and salinity of about 0.01 °C and 0.01, respectively, with a corresponding increase in $\sigma_\Theta$ of approximately 0.005 kg m$^{-3}$ (Fig. 6). The increase started at potential temperatures and salinities of 12.91 °C  and 38.49, respectively. These values were even higher than those observed in previous investigations (Table 2), which detected a continuous increase in potential temperature, salinity, and layer thickness starting in winter 2004/2005 (Schroeder et al.,

2008; Schroeder et al., 2016).

**4 Short-Term Variability**

Due to various repetitions of CTD casts at fixed positions, the REP14-MED data set is perfectly suited for studying short-term variability. Since most CTD casts covered only the upper water column, the variability estimations were restricted to MAW and LIW. Therefore, the differences between the CTD sensors were not considered. They were negligible with respect

to the natural variability of the system.

In order to study variability of water mass characteristics in the region, a grid of stations was designed and for each grid point all profiles obtained from various instruments in close vicinity were gathered to investigate short-term temporal changes at a fixed location. The grid consisted of 259 stations, 107 of them almost matched the positions of the CTD stations

which were 6 nmi apart (Fig.1). The other 152 stations were located along the glider tracks between 39°21' N and 40°3' N and between 7°15' E and 8°18' E with a distance of 3 nmi in the east-west and 6 nmi in the north-south direction. The glider as well as the ScanFish data were split into separate, alternating dive and climb profiles. For each profile a geographical position was determined by calculating the mean position during the respective dive and climb period. All CTD, UCTD, CTD chain and ScanFish profiles within 2 nmi of a grid station were collected. The glider profiles within a zonal and

meridional distance of 1 and 2 nmi, respectively, from a grid station were assigned to this position. All CTD chain, ScanFish and glider profiles recorded in quick succession and allocated to the same grid station were replaced by the corresponding



mean profiles. Between 0 and 13 profiles were allocated to each of the 259 grid stations (Fig. 12 a). No profile was assigned to 13 grid stations which were located at the eastern edge of the glider tracks. On 37 grid stations only one profile was collected and 113 grid stations had 4 or more profiles assigned. The grid stations with more than one profile covered time periods between the first and last profile of 0.18 to 16.26 days (Fig. 12 b). The short time periods were recorded at grid

stations close to the turning points of the glider tracks. Time periods of 8 days and more were allocated to 139 grid stations.

To eliminate variations resulting from internal waves, temperature and salinity profiles were interpolated on $\sigma_\Theta$ levels with a step size of 0.02 kg m$^{-3}$, which was above the observed accuracy. At the $\sigma_\Theta$ levels of the cores of MAW and LIW, mean temporal values of potential temperature and salinity as well as corresponding standard deviations and temporal gradients

based on a linear least square fit were calculated (Fig. 13). The same calculations were carried out for the mean potential temperature and salinity values within the $\sigma_\Theta$ ranges of MAW and LIW (not shown). Both calculations showed very similar spatial distributions. Within MAW (Fig. 13, left panel), the highest salinities were observed close to the shore and in the anticyclonic eddy observed in the southern part of the trial area. The strongest variability occurred in close proximity to this eddy, as well as in the southwestern part of the domain, where a mean temporal decrease in salinity was observed. This was

the area where unusually low saline water entered the region at the surface towards the end of the experiment. Farther north, the salinity variability was low. Occasional temporal increases in salinity were observed at those stations, where the corresponding time periods of the assigned profiles were short (Fig. 12 b). The LIW core covered most of the trial area west of the shelf (Fig. 13, right panel). Less saline water was observed only in the northwestern part of the area, separated by a frontal region. The highest variability occurred at this front, with a temporal increase in salinity observed. A comparison

between the CTD measurements of *Alliance* and *Planet* separated by about 8 days confirmed a northward movement of the front during the experiment. Temporal variability and gradients were comparatively low in all other regions. Corresponding calculations of variability and temporal gradients for potential temperature (not shown) yielded similar results.

## 5 Circulation

Underway current measurements were carried out with a vessel-mounted 75 kHz ADCP on *Alliance* and a 150 kHz ADCP

on *Planet*. Due to severe functional problems all *Planet* ADCP data before 10 June were discarded. The later current measurements only had sufficient quality between 100 and 150 m. There, they agreed well with the simultaneously recorded ADCP data of *Alliance* (Knoll et al., 2015b). Due to the working schedule of *Alliance*, CTD profiles of the zonal transects were often not obtained consecutively, hence geographical maps of the current data often showed a particularly strong mixture of spatial and temporal changes.

Three periods of continuous current measurements at 125 m are presented in Fig. 14. Northward currents were mainly observed offshore, while a southward flow often occurred over the continental slope. At the end of the experiment, the



ADCP measurements at 125 m depth showed southward currents in the southwest of the domain (Fig. 14 c). The geographical position, depth and time of these southward currents coincided with the TML water mass that appeared in this region. Although in general, the TML water moved northward, this water mass may have been embedded in an anticyclonic eddy, with southward currents at its eastern edge. The anticyclonic eddy observed in the southern part of the trial area was

evident in the upper ADCP measurements of *Alliance* (Russo et al., 2015, Borrione et al., 2015). The characteristics of this eddy (Fig. 9) were different from those expected in an AE with low salinities in the center, or in a newly generated SE, since the eddy was limited to the upper 100 m. The origin of this eddy should be further investigated. Like during the MedGOOS 3 observations (Ribotti et al., 2004), there was no clear indication of an AE or SE in the REP14-MED data set. The anticyclonic movement of the WIW eddy was observed in the vessel-mounted ADCP measurements of *Alliance* (Russo et

al., 2015). The general northward movement of this eddy, as well as of the TML water observed further south (Fig. 10), matched the northward currents recorded offshore by the ADCPs (Fig. 14). The path and history of the observed WIW eddy, as well as of the TML water, should be explored in more detail.

Higher salinities were found on the shelf when compared with those observed offshore (Fig. 8). The wind measurements of

*Planet,* as well as the NOAA FNMOC predictions, showed very calm weather conditions with wind speeds often below 5 m s$^{-1}$ from northerly to westerly directions. The wind slightly increased on 15 and 16 June but immediately calmed down again. Isopycnals in the upper water column showed a slight rise towards the shelf indicating light upwelling conditions and a southward geostrophic current. However, in this case deep saltier water was forced onto the shelf, rather than fresher water from the MAW salinity minimum layer, as was observed in MedGOOS 3 (Ribotti et al., 2004). The depth of the isopycnals

directly on the shelf often decreased again towards the coast especially observed in the southern zonal transects (Fig. 7), indicating a northward flow.

Geostrophic currents and transports were calculated based on the CTD transects obtained by *Alliance* and *Planet* for the water masses of MAW ($\sigma_\Theta \leq 28.72$ kg m$^{-3}$) and LIW ($28.72 < \sigma_\Theta$ [kg m$^{-3}$] $\leq 29.1$) (Fig. 15). A level of no motion (LNM) was

assumed at 1000 dbar, which was within the upper WMDW and the maximum depth level of most of the CTD profiles. In shallower areas, the reference layer was set to the deepest value close to the bottom. For comparison, the LNM was also set to the isopycnal surface of $\sigma_\Theta = 29.1$ kg m$^{-3}$, the boundary between LIW and WMDW (not shown). The geostrophic transports calculated with the different LNMs were similar, and the largest differences occurred within the meridional transports of LIW at the western boundary of the domain. Due to the rise of the $\sigma_\Theta = 29.1$ kg m$^{-3}$ isopycnal (Fig. 11 c), those transports

were slightly smaller compared to the ones calculated with a LNM at 1000 dbar.

Within MAW, the meridional transports on the western side of the trial area, as well as over the Sardinian shelf, headed north, while over the continental slope mainly southward transports were observed (Fig. 15 a, c). This matched the vessel-mounted ADCP measurements. The magnitude of the southward transport across each transect was about -0.15 Sv. The





western extent of this southward flow reached areas with bottom depths of more than 2000 m in the northern CTD transects. In the south it was restricted to regions with bottom depths between 200 and 1000 m. This southward flow resembled the WSC described by Olita et al. (2013), and the northward current in the west could be part of the Algerian Gyre (Testor and Gascard, 2005). The mean net transports across the zonal transects were mainly heading towards the north. They decreased

from 0.38 Sv in the south to almost zero across the northern four transects. Measurements along the meridional transects were not synoptic, because the ship's route was mostly in the east-west direction, following the zonal transects shown in Fig. 1. Nevertheless, transport estimates were also calculated along meridional transects, as they could give additional insights into the MAW and LIW transports in the region. The zonal transports (Fig. 15 e) showed mean westward transports around -0.2 Sv west of 7°30' E and almost zero net transports in the eastern part of the trial area. The strong east- and

westward transports in the southern domain at about 39°21' N and 30°15' N, respectively, as well as the nearby north- and southward currents, related well with the anticyclonic eddy described earlier (Russo et al., 2015, Borrione et al., 2015).

Within the LIW layer, northward transports prevailed over the whole trial area (Fig. 15 b, d). Locally observed southward transports did not present a continuous flow, but in conjunction with the zonal transports, probably indicated anticyclonic

eddies in this layer (e.g. the southern and the WIW eddy). The transport rates over the shelf were negligible, since there the LIW covered only a small part of the water column, and the LNM was set to the bottom. Like in the MAW, the strongest transports to the north in the LIW occurred offshore. Only for the northernmost transects, the maximum northward flow took place over the continental slope. This fitted to the hydrographic distribution of LIW described earlier. The net northward transport of about 0.6 Sv across the southern transect declined to 0.3 Sv across the northern one (Fig. 15).

**6 Summary and Conclusions**

The hydrography and circulation west of Sardinia observed in June were presented and compared with previous knowledge. Since the hydrographic survey was based upon different CTD instruments, a sensor comparison was initially carried out. Because $\sigma_\Theta$ in the WMDW was nearly constant, small but nevertheless significant differences could be detected. Despite careful calibrations of the various sensors before the cruise, small but systematic differences in $\sigma_\Theta$ of about $\pm 0.01$ kg m$^{-3}$

were observed between some instruments, which was slightly above the expected nominal accuracies. Looking at profiles deeper than 2000 dbar, identified conductivity as the cause of the differences between the Sea-Bird CTD sensors of *Planet* and *Alliance*. Systematic differences between the other deep profiling instruments (ARGO float, Seagliders, and Slocum gliders) appeared in the $\sigma_\Theta$ field, but the cause of this could not be identified. In the upper ocean, the differences between the sensor platforms were negligible compared to the natural variability. However, for studying density variations at greater

depth, the shifts between the different sensors have to be taken into account. When planning to use different instruments in future studies to investigate small variations, it is important to pay attention to possible significant sensor discrepancies.



The characteristics of the different water masses observed during REP14-MED were presented and compared with previous results in the area. The most interesting features within the MAW, with $\sigma_\Theta < 28.72$ kg m$^{-3}$ and salinity minima of about 37.93 at 50 dbar, were a 50 km wide, anticyclonic eddy of unknown origin in the southern part of the domain, the appearance of unusually low saline surface water entering the area from the south at the end of the experiment, and a zonal salinity gradient

with highest salinity values on the shelf. This gradient seemed quite common, either due to coastal upwelling or salinity reduction offshore caused by AEs. Unlike many other observations in this area (Ribotti et al., 2004; Testor and Gascard, 2005), no AEs or SEs were observed. Overall, the salinity in MAW in the Sardinian Sea has increased compared to previous investigations (Sparnocchia et al., 1994, Sorgente et al., 2003).

Further below, WIW was observed in an anticyclonic, northward moving eddy with potential temperature and salinity values below 12.9 °C and 38.3, respectively, at 250 dbar. No previous observation of a WIW eddy in the Sardinian Sea with such low temperatures and salinities has been documented in the past. Slightly warmer and shallower TML water coming from the south was only recorded at the end of the cruise. During the MedGOOS experiments, TML was observed all over the Sardinian Sea (Sorgente et al., 2003, Ribotti et al., 2004). The path and history of the observed WIW eddy, as well as of the

TML water, should be explored in more detail.

The LIW layer located at potential density anomalies between $28.72 < \sigma_\Theta$ [kg m$^{-3}$] $< 29.10$ covered the whole trial area from the Sardinian coast to 7°15' E, but the thickness of the layer decreased towards the west. The LIW vein was tied only to the continental slope north of 40° N, as was observed for the whole area in previous measurements (Sorgente et al., 2003). The

LIW core exhibited mean potential temperatures and salinities of 13.91 °C and 38.70, respectively. These values were higher than the historical data in the Sardinian Sea (Sparnocchia et al., 1994), likely due to the transfer of the EMT signal to the WMED (Roether et al., 2014).

The WMDW was divided into two parts, with larger vertical gradients in the upper water column, and smaller variations in

the lower water column (below 1500 dbar). The potential temperature and salinity values of the lower WMDW of 12.9 °C and 38.38, respectively, were in good agreement with earlier investigations (La Violette, 1994; Benzohra and Millot, 1995; Testor and Gascard, 2005). The rise in temperature and salinity in the BW due to the WMT was observed in the deep CTD profiles. The potential temperature and salinity values of > 12.91 °C and > 38.49, respectively, confirmed the continuous increase since winter 2004/2005 (Schroeder et al., 2008; Schroeder et al., 2016). In general, the CTD measurements

confirmed the steady increase of the heat and salt content in the different water masses of the WMED observed since the last decades. The data set might contribute to further investigations addressing the long-term evolution on hydrography in the WMED, where considerable changes occurred after EMT and WMT.



Due to various repetitions of CTD casts at fixed positions, the data set was perfectly suited for studying short-term variability in the upper water column. In the MAW, the strongest variability was observed around the southern eddy, as well as in the southwestern part of the region, where unusually low saline surface water entered the area after 17 June. In the LIW, the highest variability occurred at the front in the northwest, moving northward over the course of the experiment.

The direct current measurements in the upper ocean obtained with vessel-mounted ADCPs reflected the occurrence of anticyclonic eddies (e.g. the southern and WIW eddy). Northward currents were mainly observed offshore, while a southward flow often occurred over the continental slope. The relatively calm wind conditions induced only very light upwelling, increasing the salinity on the shelf.

Geostrophic currents and transports were calculated across the CTD transects assuming a LNM at 1000 dbar. In general, the results matched the ADCP observations. Within the MAW, the meridional transports offshore and over the shelf headed north, while over the continental slope mainly southward transports of about -0.15 Sv were observed, resembling the WSC described by Olita et al. (2013). The mean net transports across the zonal transects were mainly heading towards the north,

15 decreasing from 0.38 Sv in the south to almost zero in the north. Within the LIW, northward transports of about 0.6 Sv in the south primarily occurred offshore, decreasing to 0.3 Sv in the north, where it was located over the slope, corresponding to the observed spreading of LIW. There was no continuous southward flow over the slope but occasional southward currents were probably often part of anticyclonic eddies in the LIW. The observed northward transport rates west of Sardinia were about half of the estimated 1 Sv LIW entering the WMED.

*Author contributions.* Michaela Knoll was responsible for the collection and processing of CTD, CTD chain, UCTD, and ADCP data on *Planet*. Ines Borrione and Aniello Russo collected and processed the CTD, ScanFish, and ADCP data of *Alliance* and the CMRE Slocum gliders. Andreas Funk processed the data of the WTD71 glider. The Seaglider data were
25 processed by Michael Hemming and Bastien Queste, supervised by Jan Kaiser. Reiner Onken was the coordinator of the experiment and chief scientist on Alliance, while Heinz-Volker Fiekas was chief scientist on *Planet*. Michaela Knoll prepared the manuscript with contributions of all co-authors.

*Acknowledgements.* The authors want to thank all crew members of the research vessels *Alliance* and *Planet* for helping to
30 collect such an extraordinary data set. Thanks go to all scientists and technicians involved in the field campaign and data processing. Special thanks go to Pierre-Marie Poulain for providing the ARGO float data, and to Karen Heywood, the principal investigator from UEA, and Gareth Lee, head of the UEA Glider Facility, for providing the Seagliders. Jan Kaiser, Karen Heywood, and Michael Hemming were supported by funding from the Defence Science and Technology Laboratory



(DSTL, UK) in close co-operation with Direction générale de l'armement(DGA, France). CMRE activities, including *Alliance*, were supported by the NATO Allied Command Transformation (ACT) through project SAC000404.

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



**Table 1.** Abbreviations of water masses, features and areas used in this study. The acronyms mainly correspond to the recommendations of the round table session on Mediterranean Water Mass Acronyms at the 36th CIESM (Commission Internationale pour l'Exploration Scientifique de la mer Méditerranée) Congress in Monte Carlo in September 2001 (http://ciesm.org/events/RT5-WaterMassAcronyms.pdf).

| Acronym | Water mass, Feature, Area |
|---------|---------------------------|
| AE | Algerian Eddy |
| BW | Bottom Water |
| EMT | Eastern Mediterranean Transient |
| LIW | Levantine Intermediate Water |
| MAW | Modified Atlantic Water |
| SE | Sardinian Eddy |
| TML | Temperature Minimum Layer |
| WIW | Winter Intermediate Water |
| WMDW | Western Mediterranean Deep Water |
| WMED | Western Mediterranean |
| WMT | Western Mediterranean Transition |
| WSC | Western Sardinian Current |



**Table 2.** Ranges and core characteristics of the parameters pressure, potential density anomaly ($\sigma_\Theta$), potential temperature ($\Theta$), and salinity (S) of water masses in the WMED obtained from literature and the REP14-MED data set. Specifications in brackets refer to core values. Italic indications correspond to TML characteristics.

| Water mass characteristic origin | Pressure range (core) [dbar] | $\sigma_\Theta$ range (core) [kg m$^{-3}$] | $\Theta$ range (core) [°C] | S range (core) | Reference study where values are measured or just mentioned |
|---|---|---|---|---|---|
| MAW low-salt Strait of Gibraltar | 0-140 (50) | <28.72 (28.19) | 13.6-24.2 (15.19) (summer 17.71) | 37.15-38.3 (37.93) (summer 37.61) | REP14-MED data Sparnocchia et al., 1994 |
| | 0-200 (summer:50-70) | | | 36.5-37.5 | La Violette, 1994 |
| | 0-150 | | 13.5-23 | 36.5-38.0 | Benzohra and Millot, 1995 |
| | 0-150 | | 13.7-15.2 | 37.2-38.2 | Sorgente et al., 2003 |
| WIW / *TML* cold/low-salt / *cold layer* northern WMED, not formed every year / *influenced by WIW* | 170-400 (250) | (28.99) | (<12.9) | (<38.3) | REP14-MED data |
| | *120-270 (200)* | *(28.86-28.89)* | *(<13.25)* | *(38.24-38.30)* | |
| | (150-200) | | (12.35) (12.65-13.2) | (38.3) (38.3) | Benzohra and Millot, 1995 |
| | *150-250* | | *13.3-13.5* | | Sorgente et al., 2003 |
| | 200->360 | | (12.9) | (38.20) | Allen et al., 2008 |
| | *(<300)* | | *(<13.5)* | *(38.2)* | |
| | | 28.8-28.9 | <13 | <38.3 | Vargas-Yáñez et al., 2012 |
| | | | 11.5-13.0 | 37.7-38.3 | Juza et al., 2013 |
| LIW saline Strait of Sicily | 80-740 (402) | 28.72-29.1 (29.06) | 13.2-14.2 (13.91) (summer 13.51) | 38.24-38.74 (38.70) (summer 38.58) | REP14-MED data Sparnocchia et al., 1994 |
| | 200-800 | | | 38.45-38.75 | La Violette, 1994 |
| | (300-500) | | (13.2-14.0) | (38.5-38.7) | Benzohra and Millot, 1995 |
| | 300-800 | | 13.6-14.0 | 38.64-38.70 | Sorgente et al., 2003 |
| | 300-400 | | 13.2-14.0 | 38.5-38.7 | Juza et al., 2013; Juza et al., 2015 |
| | 200-500 | 29.03-29.10 | | (38.68-38.70) | Bosse et al., 2015 |
| WMDW $\sigma_\Theta$ changes small, monotonic decrease of $\Theta$ and S winter formation in NW Mediterranean | upper 540-1500 lower 1500-2300 | 29.10-29.11 29.11-29.117 | 12.91-13.70 12.91-12.90 | 38.48-38.68 38.48-38.485 | REP14-MED data |
| | 800-3000 | | 12.75-12.90 | 38.4-38.48 | La Violette, 1994 |
| | >600 | | 12.75-12.90 | 38.42-38.47 | Benzohra and Millot, 1995 |
| | | | 12.8-12.9 | 38.48 | Testor and Gascard, 2005 |
| | >2000 | 29.1 | ≤ 12.83 in 2004 | ≤ 38.45 in 2004 | Schroeder et al., 2008 |
| | >1000 | 29.1 | 12.85-12.89 | 38.48-38.50 | Bosse et. al, 2015 |
| BW developed after WMT increase of $\Theta$, S, $\sigma_\Theta$ near bottom same origin as WMDW | >2300 | >29.118 (increase 0.005) | >12.91 (increase 0.01) | >38.49 (increase 0.01) | REP14-MED data |
| | >3000 | | | | La Violette, 1994 |
| | >2000 | | 12.85 in 2005 | 38.46 in 2005 | Schroeder et al., 2008 |
| | >2000 | | 12.85-12.88 in 2006 | 38.455-38.47 in 2006 | Schroeder et al., 2008 |
| | | >29.11 in 2010 | | | Schroeder et al., 2016 |
| | 1900 | 29.119 | 12.90 in 2013 | 38.485 in 2013 | Schroeder et al., 2016 |





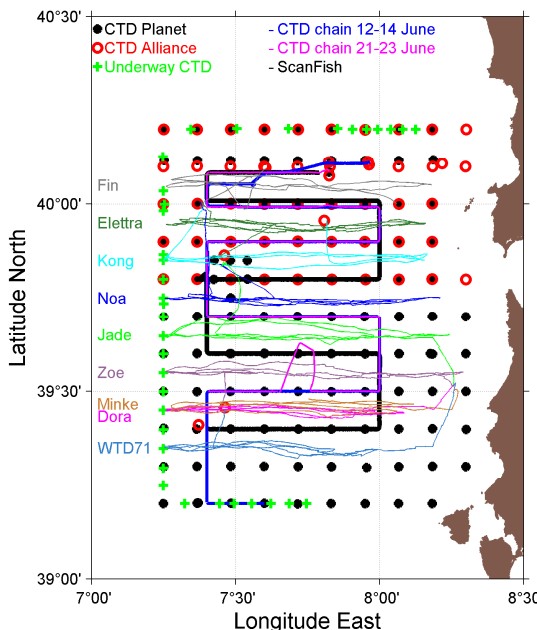

**Figure 1.** CTD data sets obtained with different instruments. Glider names are specified next to their tracks.

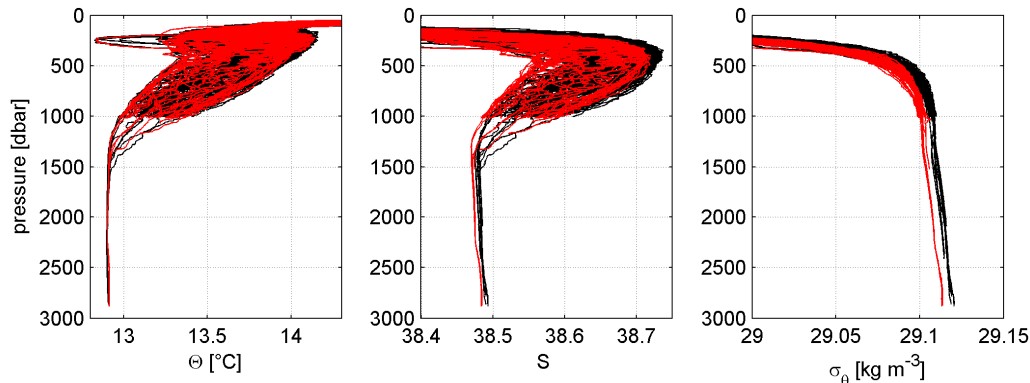

**Figure 2.** Potential temperature (Θ), salinity (S) and potential density anomaly (σ_Θ) profiles of Sea-Bird CTD casts obtained by *Planet*
5   (black) and *Alliance* (red) with focus on the lower water column.



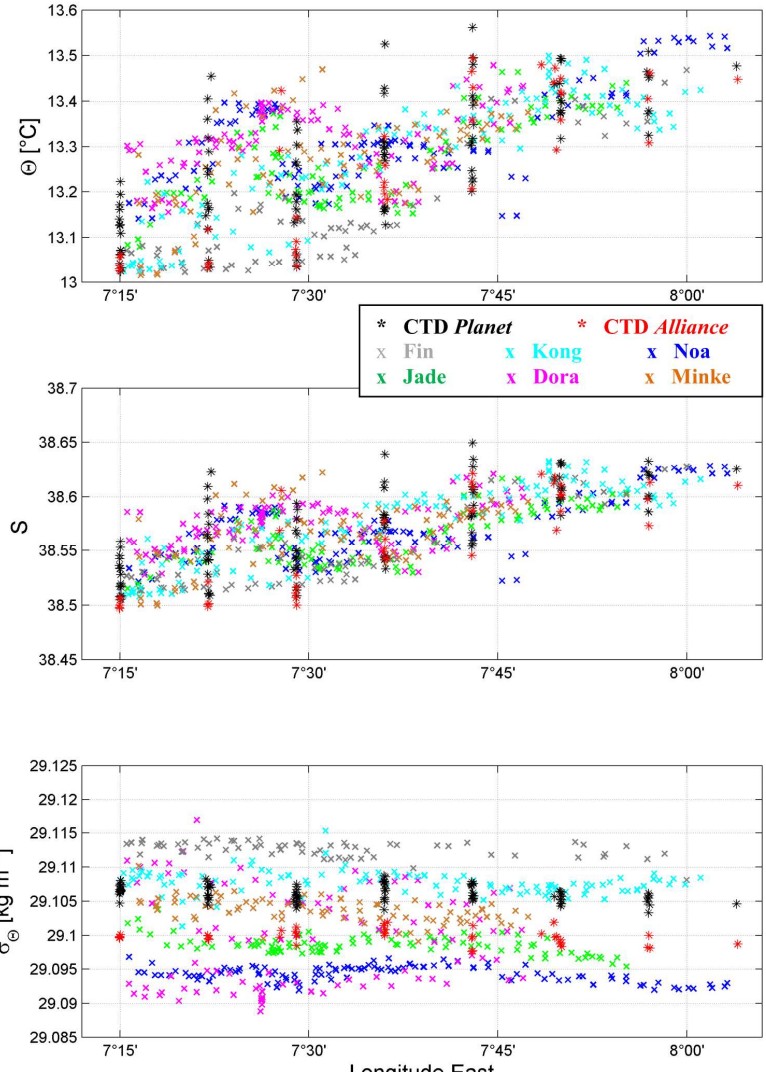

**Figure 3.** Potential temperature (Θ), salinity (S) and potential density anomaly (σ_Θ) at roughly 900 dbar versus longitude of all CTD and glider profiles available at that pressure level. *Planet* covered all latitudes of the trial area, *Alliance* observed in the northern part, and the gliders surveyed at selected latitudes given in Fig. 1.




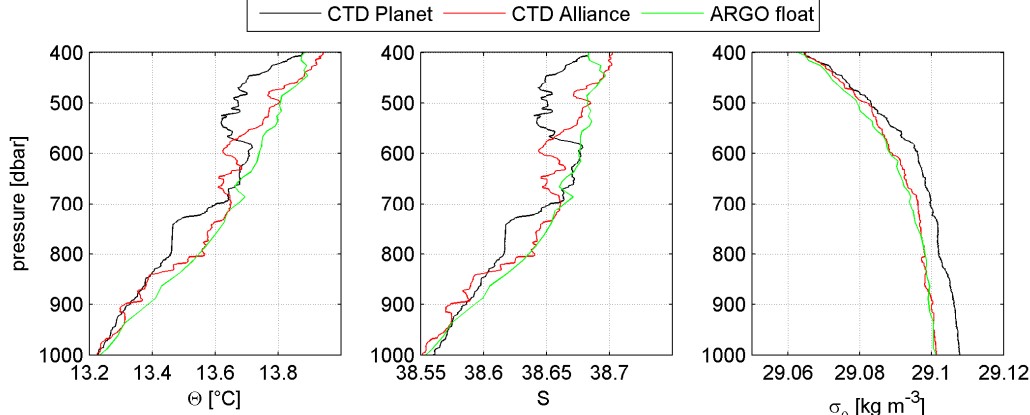

**Figure 4.** Potential temperature (Θ), salinity (S) and potential density anomaly (σΘ) profiles obtained from ARGO float 6901836 (14 June) and nearby profiles of *Planet* (17 June) and *Alliance* (11 June).

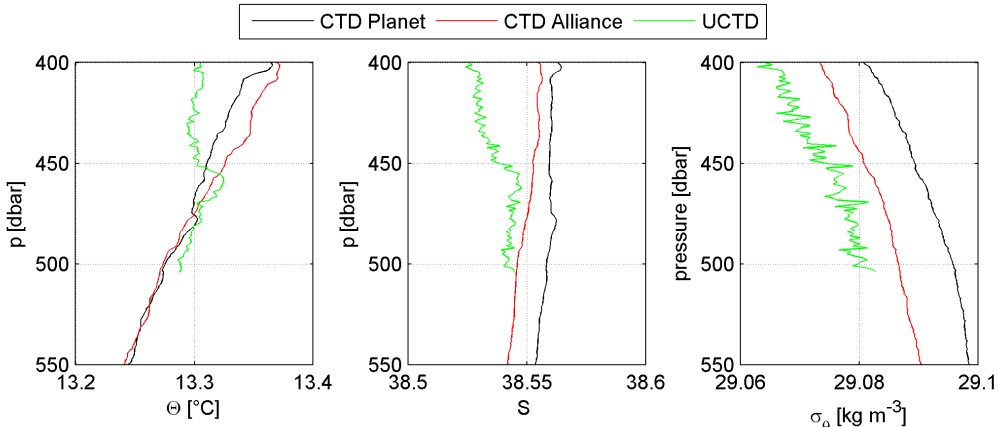

5    **Figure 5.** Potential temperature (Θ), salinity (S) and potential density anomaly (σΘ) profiles of the Underway CTD (UCTD) (15 June) and nearby profiles of *Planet* (18 June) and *Alliance* (9 June).





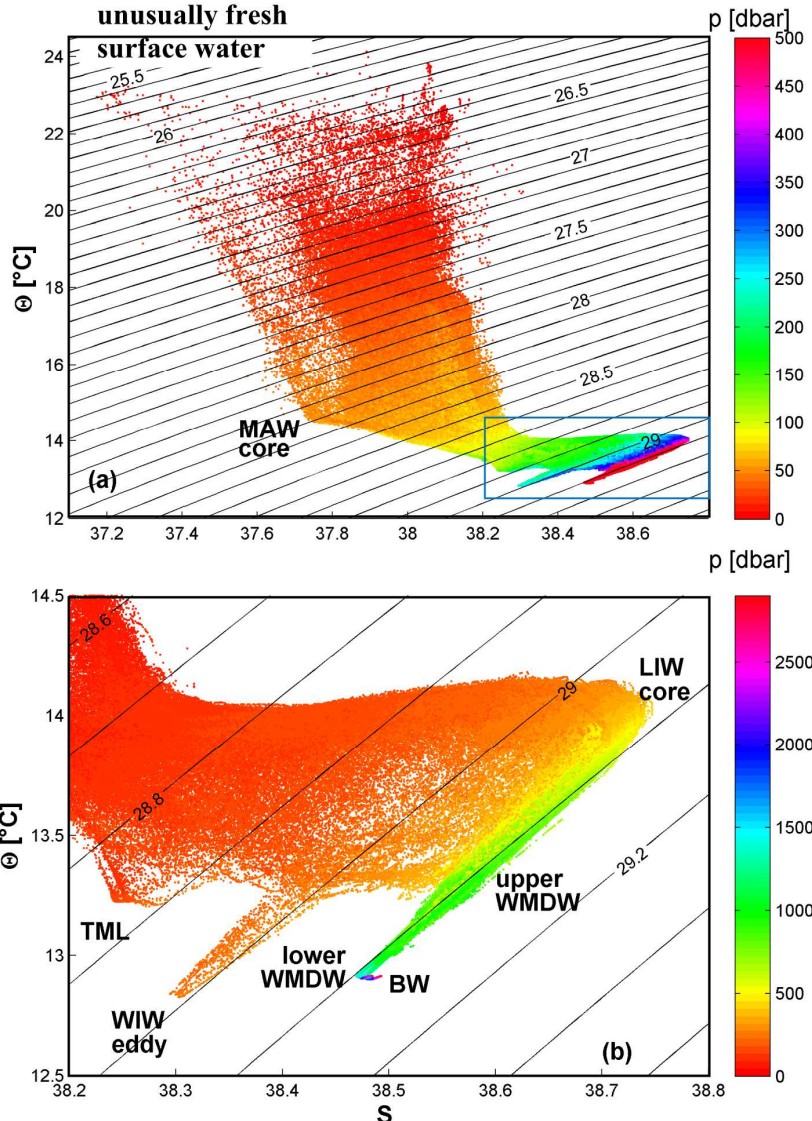

**Figure 6.** Potential temperature/salinity (Θ/S) diagram with contours of potential density anomaly (σ_Θ) (black) of all CTD casts and gliders. Diagrams are presented for the **(a)** whole water column with coloured pressure in the upper 500 dbar and unicolour below (blue box indicates the range shown in Fig. 6 b), and **(b)** lower water column with coloured pressure between 0 and 2900 dbar.





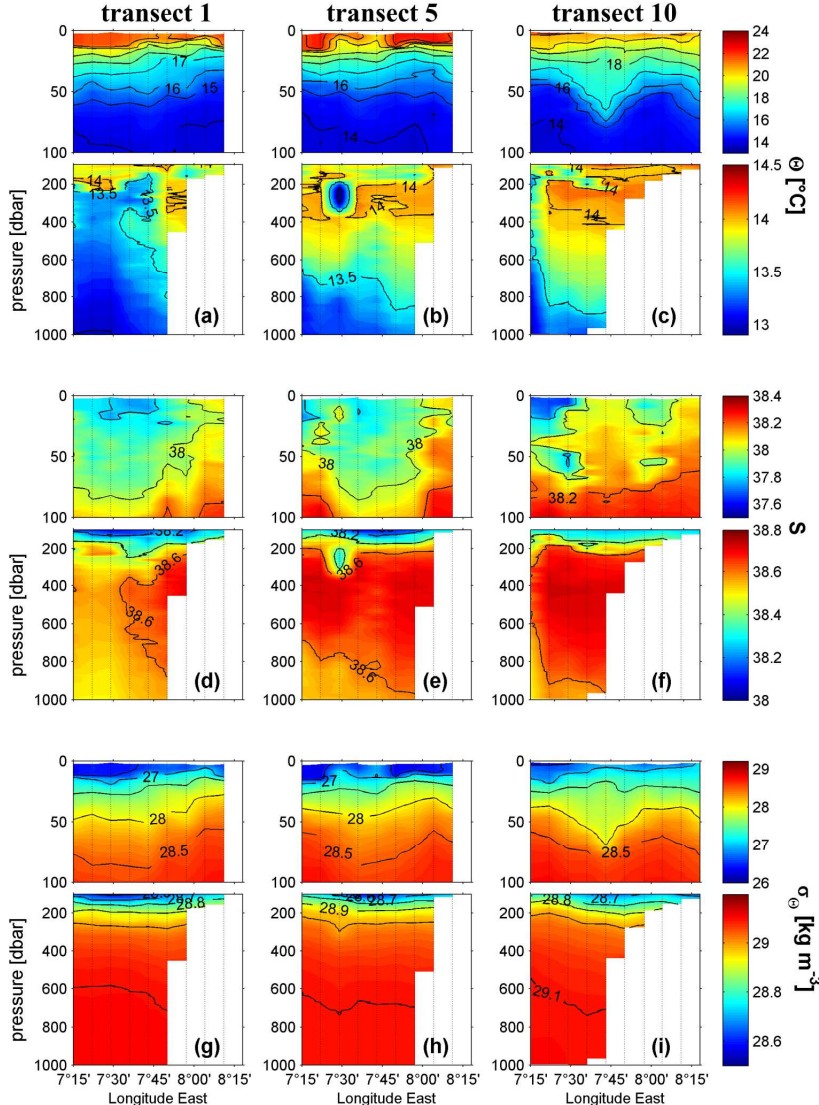

**Figure 7.** Potential temperature (Θ), salinity (S), and potential density anomaly (σ_Θ) along the zonal **(a, d, g)** transect 1 (40°12' N), **(b, e, h)** transect 5 (39°48' N), and **(c, f, i)** transect 10 (39°18' N) in the upper 100 and 1000 dbar based on *Planet* CTD casts (dotted lines) given in Fig. 1.




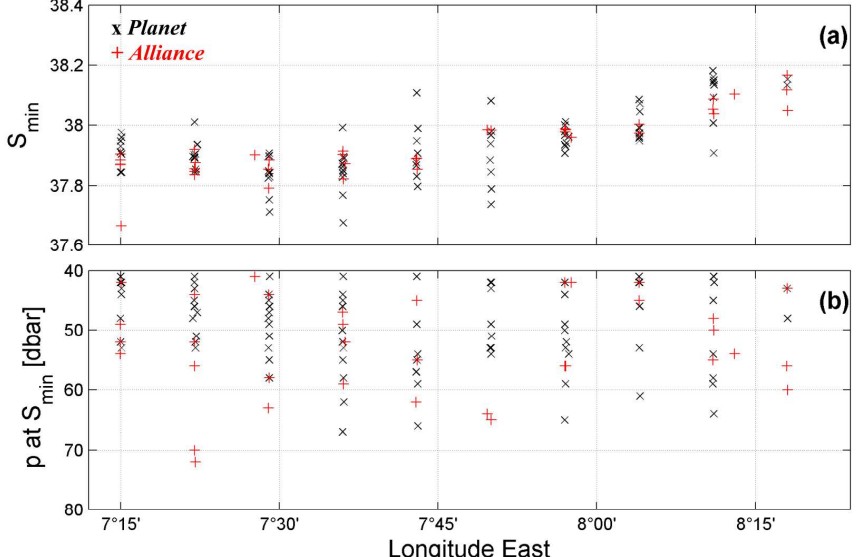

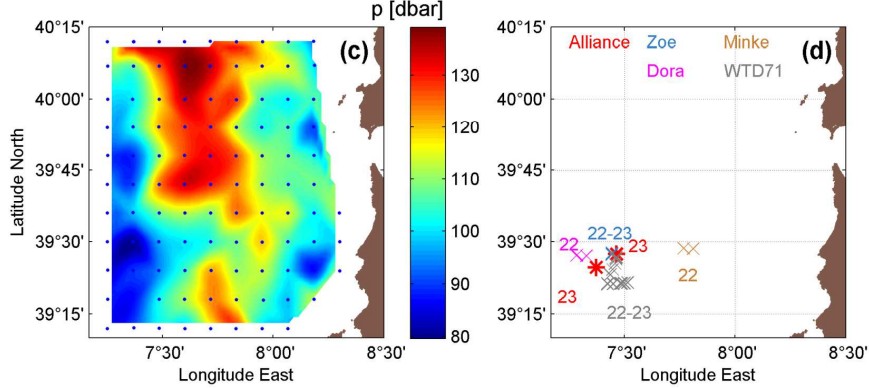

**Figure 8.** Characteristics of MAW, **(a)** salinity minimum ($S_{min}$) within 40 to 80 dbar of all *Planet* (**x**) and *Alliance* (+) CTD casts, **(b)** corresponding pressure (p) levels of $S_{min}$, **(c)** pressure level of lower boundary of MAW ($\sigma_\Theta = 28.72$ kg m$^{-3}$) based on *Planet* CTD casts, **(d)** Positions and June dates of all CTD and glider profiles with measurements of unusually low saline surface water (S < 37.32 within 0 to 50 dbar).



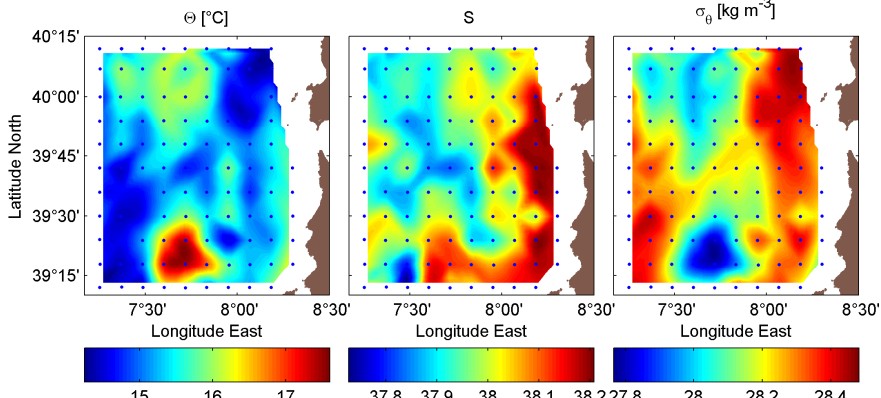

**Figure 9.** Potential temperature (Θ), salinity (S), and potential density anomaly ($\sigma_\Theta$) distribution at the pressure level of 50 dbar based on CTD casts (blue dots) obtained by *Planet* from 8-18 June.

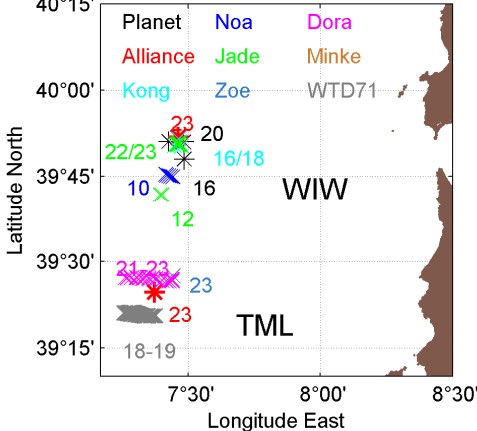

**Figure 10.** Positions and June dates of all CTD and glider profiles with WIW and TML observations, respectively (T < 13.3 °C and S < 38.33 between 100 and 350 dbar).





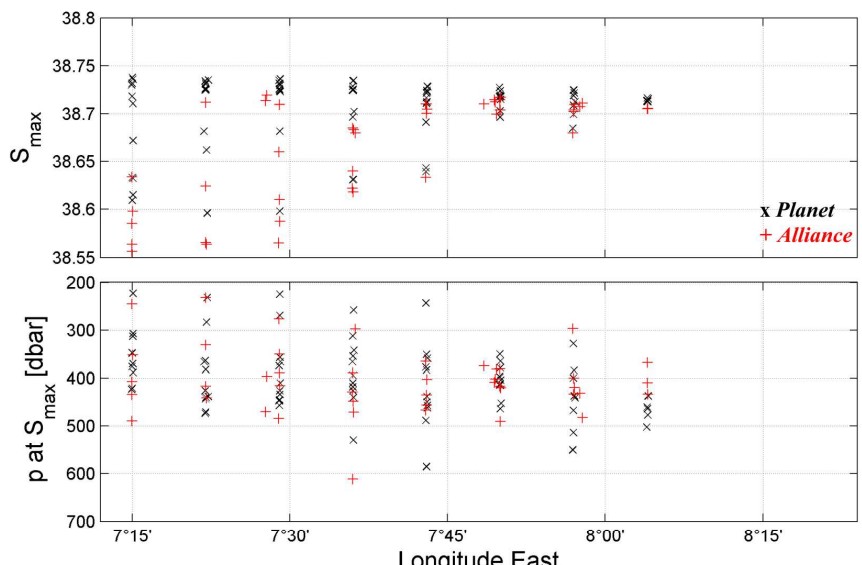

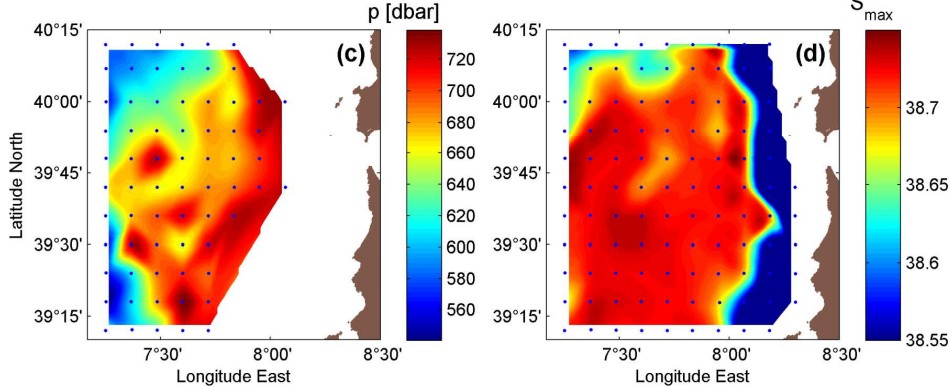

**Figure 11.** Characteristics of LIW, **(a)** salinity maximum ($S_{max}$) of at least 400 dbar deep CTD casts of *Planet* (**x**) and *Alliance* (**+**), **(b)** corresponding pressure level of $S_{max}$, **(c)** pressure level of lower boundary of LIW ($\sigma_\Theta = 29.1$ kg m$^{-3}$) based on *Planet* CTD casts (blue dots), **(d)** maximum salinity ($S_{max}$) > 38.55 based on *Planet* CTD casts (blue dots).




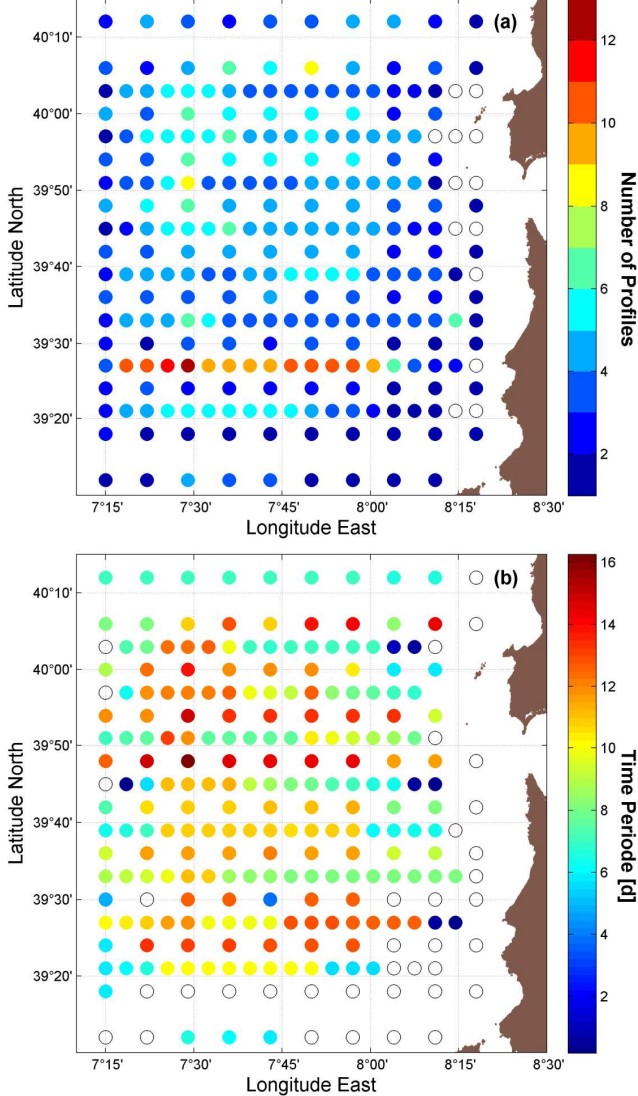

**Figure 12. (a)** Number of profiles gathered for each grid station (stations with no assigned profiles in white), and **(b)** time period in days between the first and last profile assigned to each grid station (stations with only one profile in white).



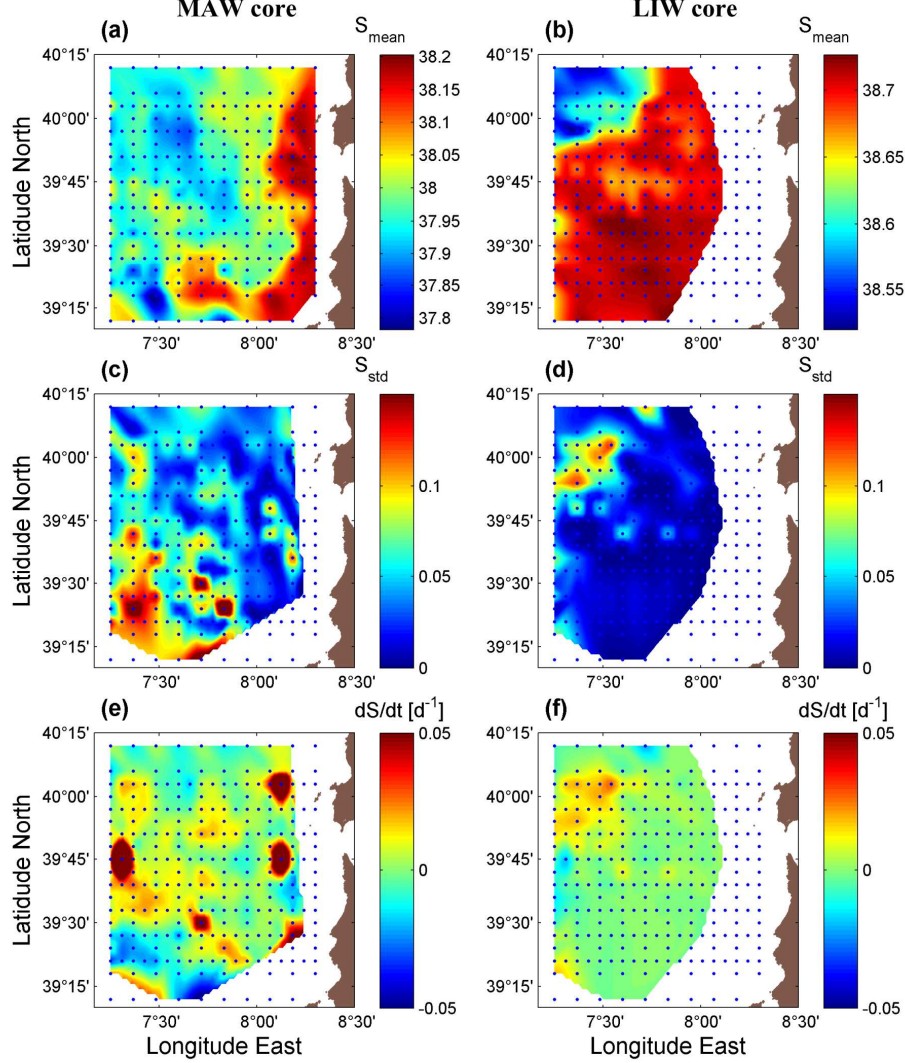

**Figure 13.** Mean salinity ($S_{mean}$), standard deviation ($S_{std}$), and linear temporal change in salinity per day (dS/dt [$d^{-1}$]) in the cores of **(a, c, e)** MAW ($\sigma_\Theta$=28.2 kg m$^{-3}$) and **(b, d, f)** LIW ($\sigma_\Theta$=29.06 kg m$^{-3}$) based on all collected data of the station grid (blue dots).




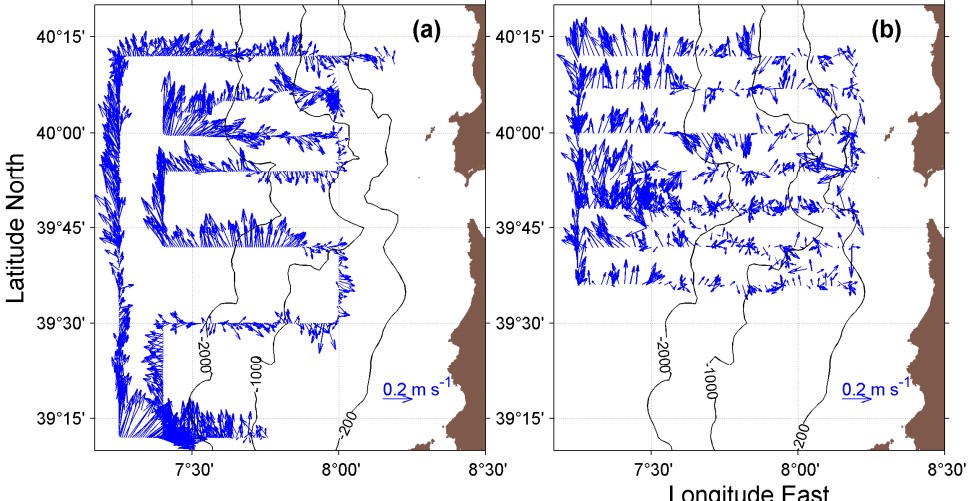

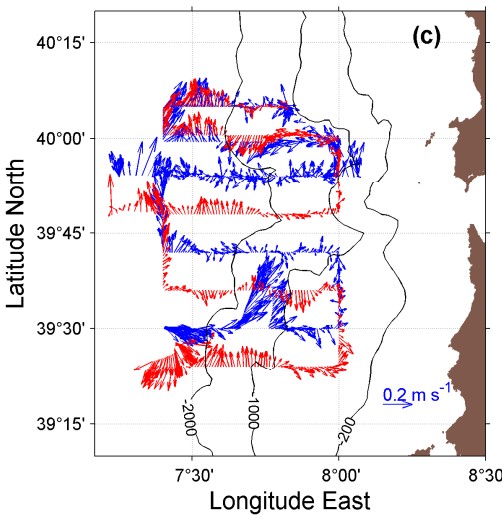

**Figure 14.** ADCP current vectors of *Alliance* (red) and *Planet* (blue) at 125 m depth averaged over 300 s for the time periods **(a)** 11 to 16 June, first CTD-chain tow, **(b)** 16 to 20 June, northern CTD-transects, **(c)** 21 to 23 June, second CTD-chain / ScanFish tow, including the isobaths at 200, 1000 and 2000 m depth.




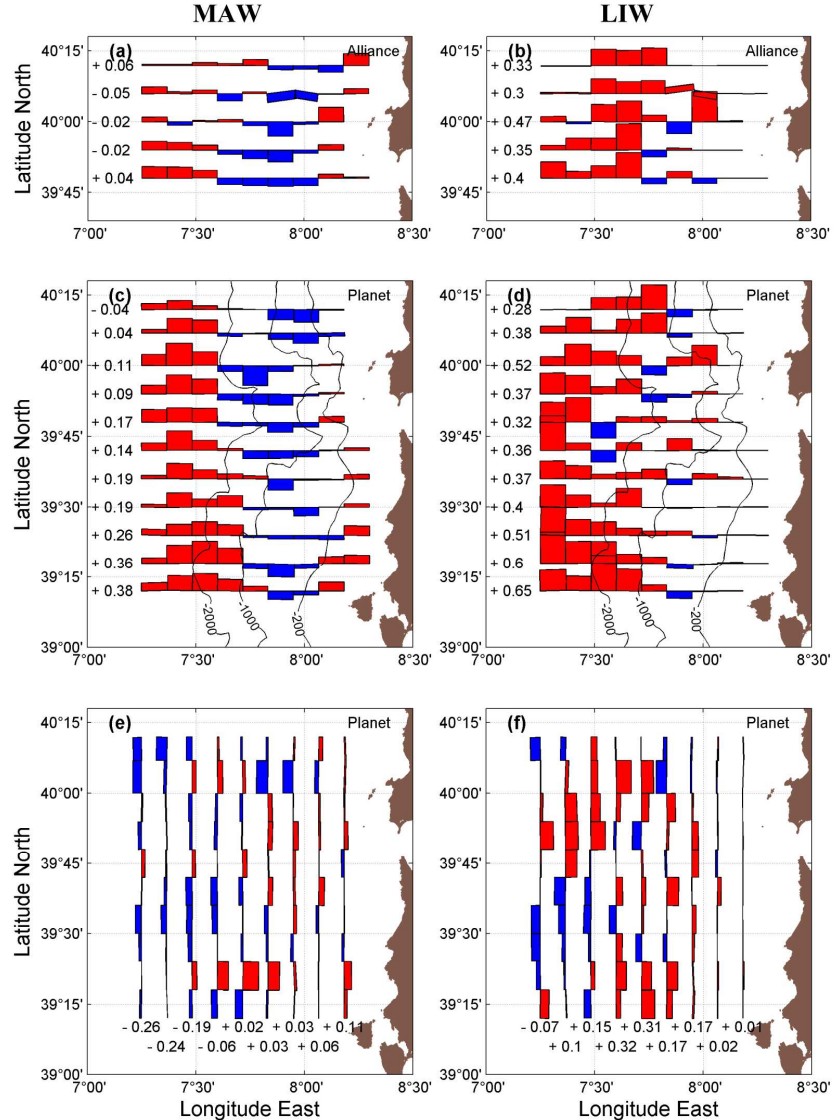

**Figure 15.** Geostrophic transports [Sv] calculated with a level of no motion (LNM) at 1000 dbar and bottom, respectively, along the CTD transects of **(a, b)** *Alliance* and **(c, d, e, f)** *Planet* within the water masses **(a, c, e)** MAW and **(b, d, f)** LIW. The net transports are noted beside each transect. Plots of the meridional transports observed by *Planet* additionally show the isobaths at 200, 1000 and 2000 m depth.