# Peer review of "Hydrography and Circulation West of Sardinia in June 2014"

_Ocean Science, 2017_

## Referee Comment (RC1) · Anonymous Referee #1 · 8 Jun 2017

general comments

The manuscript titled "Hydrography and Circulation West of Sardinia in June 2014" aims to describe short-term variations in hydrography and circulation of the upper layers west of Sardinia in June 2014. The aim is to give further updated information on the area compared with previous studies to investigate long-term changes. To do this, the authors use the huge amount of data acquired during the REP14-MED oceanographic survey, carried out for 18 days from 6 to 25 June 2014 and supported by NATO. The paper is well written and describes the results on hydrodynamic data. It does not give important updates on the hydrodynamics as it mainly agrees with previous studies without analysing a particular phenomenon in detail. But its interest is mainly on the fact that it give further information on an area that, as written by the authors, "...is scarcely

known from previous investigations". Can be acceptable for publication after minor revisions on specific comments.

specific comments

Two are the suggestions to improve the analyses in the whole manuscript: 1. it is not clear to me why to introduce the "new water" Temperature Minimum Layer (TML) in the wide water masses panorama as the same authors define it as "not an autonomous water mass, but consists of WIW modified continually on its way through the WMED." So I would suggest to use WIW anyway, even if it has characteristics that differ from other WIWs. If authors evaluate it is necessary a distinction then, like AW transforms in MAW during its path, I would suggest to define it as modified Winter Intermediate Water (mWIW). This would better avoid any confusion and let immediately understand of what you are speaking about. Modifications in the whole manuscript are then necessary.

2 two important papers, not mentioned here, should be considered in the comparison of water masses with historical data in the whole manuscript, in my opinion: the first paper is on water masses in the Sardinia Channel Bouzinac C., J. Font, C. Millot, (1999). Hydrology and currents observed in the channel of Sardinia during the PRIMO-1, experiment from November 1993 to October 1994. J. Mar. Sys., 20, 1-4, 333–355, https://doi.org/10.1016/S0924-7963(98)00074-8;

the second focuses on the study of LIW in the Sardinian Sea in 2002-2004 and completes other papers mentioned in the manuscript (Sorgente et al., 2003; Puillat et al., 2003; Ribotti et al., 2004) on the hydrodynamics in the area Puillat I., R. Sorgente, A. Ribotti, S. Natale, V. Echevin, (2006). Westward branching of LIW induced by Algerian anticyclonic eddies close to the Sardinian slope, Chemistry and Ecology, 22, S1, S293 - S305, DOI: 10.1080/02757540600670760

technical corrections

page 3, line 29: delete comma (,) before bibliographic reference page 4, line 3: change

"whole area" with "northern part of the area" as it is mentioned that it occurred above 39.6 °N (Ribotti et al., 2004) page 4, line 10: change 300 m in 400 m (see in Bouzinac et al., 1999; Puillat et al., 2006) page 9, line 15 and page 13, line 29: add Borghini et al., 2014. Its Table 2 perfectly fits with what mentioned in the sentences page 12, line 21: delete the whole line as it is repeated at page 13, line 1 page 13, line 10: delete "moving" as there are no permanent eddies in the Sardinian Sea

---

## Referee Comment (RC2) · Anonymous Referee #2 · 26 Jun 2017

**General comments**

This manuscript describes the hydrography and circulation features observed west of Sardinia during the REP14-MED multi-platform experiment. It gives an interesting snapshot of the oceanographic characteristics in the study area during the period 6-25 June 2014, but the lack of new/real results lead the authors to add too much information (about the sensors calibration/accuracy, water masses, figures, etc...). These overflow of information distract the reader and make too hard to follow a guide line. For these reasons, I suggest to simplify the structure of sections 2, 3, and 5 in order to emphasize only the most important aspects that arise from the data. I recommend the publication of the manuscript after minor revisions suggested in the detailed comments and after a general lighten of the text.

[Figure]

Detailed comments

Section 2: I suggest to reduce strikingly the comparisons between sensors because this topic was already discussed in the technical report of Knoll et al 2015 and because, as the same authors explain in the first paragraph of section 4, "the differences between CTD sensors were not considered".

Section 3. I suggest to don't repeat in the text the hydrographic characteristics already summarized in Table 2 (e.g. Line 28.... varied between 13.6 and 24 °C and between 37.1 and 38.3...etc)

Page 6, Line 30: Add a sentence to introduce/describe Fig 7 before to speak about their evidences/results.

Section 5: reduce the first paragraph (Lines 24-29);

Lines 14-21: this paragraph appears not related with the context of Currents; put it in the right context or remove it.

Fig. 1: Add bathymetry lines as in Fig 14 and specify the geographical location of transect 1, 5 10;

Figure 12: remove this figure and add other two subplots in Figure 13

---

## Referee Comment (RC3) · Anonymous Referee #1 · 7 Jul 2017

As written in Ribotti et al. (2004), the upwelling mainly occurred over 40 °N with a spreading of the cold water 50 Km offshore (page 358 and the satellite picture at page 360). That area can be defined as the northern part of the sampling area. The salinity inversion occurred over the "entire sampling field: the cross-shore transect near 39° 60'N (page 359)" and is not clearly explained as some contrasting characteristics are described by the authors. It is then difficult to define it as a classical upwelling.
* * *

---

## Author Comment (AC1) · 7 Jul 2017

we thank both referees for their helpful comments which we will address in detail at the end of the discussion period. Prior to this, we have a question to referee #1 concerning the technical correction on page 4 line 3: Change 'whole area' with 'northern part of the area' as it is mentioned that it occurred above 39.6°N (Ribotti et al., 2014). In the abstract as well as on page 359 and 361of this paper it is written that the salinity inversion occurred over the whole sampling field. Taking a quick look at MedGOOS3 CTD data we saw salinity inversions also south of 39.6°N. What makes you think it occurred only north of it?
* * *

---

## Author Comment (AC2) · 7 Aug 2017

We would like to thank both referees for their helpful comments and input, which we want to address in the following way:

Referee #1

1. Comment It is not clear to me why to introduce the "new water" Temperature Minimum Layer (TML) in the wide water masses panorama as the same authors define it as "not an autonomous water mass, but consists of WIW modified continually on its way through the WMED." So I would suggest to use WIW anyway, even if it has characteristics that differ from other WIWs. If authors evaluate it is necessary a distinction then, like AW transforms in MAW during its path, I would suggest to define it as modified Winter Intermediate Water (mWIW). This would better avoid any confusion and let immediately understand of what you are speaking about. Modifications in the whole manuscript are then necessary. Answer: the term 'Temperature Minimum Layer (TML)' was already used in some previous papers (e.g. Allan et al., 2008; Forryan et al., 2012. Turbulent mixing in the eddy transport of Western Mediterranean Intermediate Water to the Alboran Sea, JGR, 117, C09008, doi:10.1029/2012JC008284). They describe TML as a water mass which is strongly influenced by WIW but has lost its characteristic values due to mixing. Therefore, we did not create the term TML but only adopted it to distinguish between the cold water mass observed in the south and the WIW eddy. We are reluctant to introduce a new name like modified WIW (mWIW), though we agree, that this would be a suitable name and would avoid any confusions. Using the same expression WIW for both features might even increase confusion. If acceptable, we would like to stick to the term TML and replace 'influenced by WIW' by 'modified WIW' in Table 2 to make clear they have the same origin.

2. Comment: Two important papers, not mentioned here, should be considered in the comparison of water masses with historical data in the whole manuscript, in my opinion: the first paper is on water masses in the Sardinia Channel Bouzinac C., J. Font, C. Millot, (1999). Hydrology and currents observed in the channel of Sardinia during the PRIMO-1, experiment from November 1993 to October 1994. J. Mar. Sys., 20, 1-4, 333–355, https://doi.org/10.1016/S0924-7963(98)00074-8; the second focuses on the study of LIW in the Sardinian Sea in 2002-2004 and completes other papers mentioned in the manuscript (Sorgente et al., 2003; Puillat et al., 2003; Ribotti et al., 2004) on the hydrodynamics in the area Puillat I., R. Sorgente, A. Ribotti, S. Natale, V. Echevin, (2006). Westward branching of LIW induced by Algerian anticyclonic eddies close to the Sardinian slope, Chemistry and Ecology, 22, S1, S293 - S305, DOI: 10.1080/02757540600670760 Answer: thank you for pointing out the two important papers of Bouzinac et al. (1999) and Puillat et al. (2006) which we added to our study.

3. Technical Corrections:

- page 3, line 29: delete comma (,) before bibliographic reference. Answer: done

- page 4, line 3: change "whole area" with "northern part of the area" as it is mentioned that it occurred above 39.6 °N (Ribotti et al., 2004). Answer: we pointed out that the salinity inversion occurred over the whole sampling field while the upwelling mainly took place north of 40° N.

- page 4, line 10: change 300 m in 400 m (see in Bouzinac et al., 1999; Puillat et al., 2006) Answer: done

- page 9, line 15 and page 13, line 29: add Borghini et al., 2014. Its Table 2 perfectly fits with what mentioned in the sentences. Answer: done

- page 12, line 21: delete the whole line as it is repeated at page 13, line 1. Answer: done

- page 13, line 10: delete "moving" as there are no permanent eddies in the Sardinian Sea. Answer: we delete 'northward moving' in this sentence, but added another sentence concerning the northward movement of the eddy observed during REP14-Med.

Referee #2

1. Comment Section 2: I suggest to reduce strikingly the comparisons between sensors because this topic was already discussed in the technical report of Knoll et al 2015 and because, as the same authors explain in the first paragraph of section 4, "the differences between CTD sensors were not considered". Answer: we reduced the text in section 2 and eliminated former Fig. 4 and 5 to diminish the dominance of this section.

2. Comment Section 3. I suggest to don't repeat in the text the hydrographic characteristics already summarized in Table 2 (e.g. Line 28: : :. varied between 13.6 and 24 _C and between 37.1 and 38.3: : :etc). Answer: in most cases we tried to avoid
repetitions of characteristic values given in Table 2.

3. Comment Page 6, Line 30: Add a sentence to introduce/describe Fig 7 before to speak about their evidences/results. Answer: we added a sentence at the beginning of section 3 to introduce Fig. 5 (former Fig. 7).

4. Comment Section 5: reduce the first paragraph (Lines 24-29). Answer: we reduced the first paragraph of section 5.

5. Comment Lines 14-21: this paragraph appears not related with the context of Currents; put it in the right context or remove it. Answer: we removed the paragraph on weather conditions and upwelling.

6. Comment Fig. 1: Add bathymetry lines as in Fig 14 and specify the geographical location of transect 1, 5 10. Answer: We added bathymetry lines and transect numbers in Fig. 1.

7. Comment: Figure 12: remove this figure and add other two subplots in Figure 13. Answer: we changed the arrangement in Fig. 10 (former Fig. 12), but did not add them as subplots in Fig. 11 (former Fig. 13), since they do not refer to MAW or LIW like the other figure.

Please also note the supplement to this comment:
https://www.ocean-sci-discuss.net/os-2017-45/os-2017-45-AC2-supplement.pdf

**Supplement:**

[revised manuscript text omitted]